# The Quality of Surgical Instrument Surfaces Machined with Robotic Belt Grinding

**DOI:** 10.3390/ma16020630

**Published:** 2023-01-09

**Authors:** Adam Hamrol, Mateusz Hoffmann, Marcin Lisek, Jedrzej Bozek

**Affiliations:** 1Faculty of Mechanical Engineering, Poznan University of Technology, Piotrowo 3, 60-138 Poznan, Poland; 2Aesculap Chifa, Tysiaclecia 14, 64-300 Nowy Tomysl, Poland

**Keywords:** manufacturing of surgical instruments, surface quality, surface roughness, robotic belt grinding

## Abstract

Belt grinding is commonly used in the finishing of non-functional shaped surfaces of surgical instruments. Most often it is carried out manually. The subject of this article is the possibility of replacing manual belt grinding with robotic grinding. A research stand was built, the machining process was programmed, and a comparative study of manual and robotic grinding was carried out. The subject of the research were the arms of orthodontic forceps. The condition of the treated surface, defined by its structure and roughness and the geometric accuracy and the error of the shape of the arm in the selected cross-section were adopted as the comparative criteria. Research has shown that robotic belt grinding is more efficient in terms of quality and produces more consistent results than manual grinding.

## 1. Introduction

Surgical instruments are basic elements of the equipment of doctors of various specialties, especially surgeons. Since their functionality must ensure the safe and efficient performance of various medical procedures, the range of surgical instruments is very wide [1,2]. Each tool has a unique shape with predominantly curvilinear surfaces. The shape of the tool must allow access to the operated area and holding the tool securely, and must ensure its adequate strength (Figure 1). Tool surfaces that do not perform direct work, such as cutting and holding, are non-functional surfaces.

The shape of surgical tools determines the structure of their manufacturing process, which usually consists of several operations, depending on the complexity of the tool. The vast majority of surgical tools are made of forgings, manufactured in the processes of forging or stamping, either cold or hot. There are attempts to manufacture surgical tools entirely by machining or incremental methods [3]. However, the properties, especially strength, can be obtained only with the use of the traditional technologies. Therefore surgical instruments manufactured with additive methods are intended for individual applications, when there is a need to obtain their unique functionality [4].

Abrasive machining plays a large role in the typical manufacturing process of surgical tools due to their shape and surface requirements. It is used to clean the surface of forgings, to obtain the appropriate surface structure, or to give some surfaces a special shape. Shaping treatment can be carried out on conventional grinding machines or belt grinders, and cleaning of the surface on belt grinders or by using vibro-abrasive machining.

Belt grinding, mainly manual belt grinding, dominates in the implementation of all these operations. This is due to the high flexibility and versatility of manual belt grinding. As a finishing process, belt grinding not only achieves high material removal rates but can also be used to improve the surface roughness of components [5]. An experienced operator is able to remove various workpiece shape deviations after plastic processing.

The operator can individually select drive wheels of such shapes that the abrasive belt, after stretching, arranges itself in a way that enables and facilitates the processing of a given part. The operator uses various types of abrasive belts differing from each other (e.g., grain or bonding material), which makes it possible to find and select a belt with specific properties best suited to the production task at hand.

However, manual belt grinding has a number of disadvantages [5], including difficult working conditions for the human operator, which translate into a relatively high occupational risk. Difficult conditions include direct contact with the rotating drive wheel, noise and the microclimate prevailing on the grinding stand, uncomfortable body position at work, high monotony, and the need for constant concentration (Figure 2).

From a technological point of view, the weakness of grinding with abrasive belts is the difficulty in obtaining repeatability of shapes and dimensions.

Today, most parts with complex shapes are ground by hand or on multi-axis computer numeric control (CNC) machines. Due to the time-consuming and labor-consuming nature, as well as the difficult working conditions of manual grinding, CNC grinding dominates. However, it has weak points, such as the high cost of precision CNC machine tools, low flexibility and the difficulty of quickly adapting machining programs to changing the shape of the workpiece [6,7,8].

Industrial robotic belt grinding offers new possibilities for machining shaped parts. Compared to multi-axis CNC machine tools, robots are attractive because they can be flexibly adapted to the machining task. They also have a competitive price compared to CNC grinders, which makes them a cost-effective solution for processing complex elements, especially large-size elements [9].

Grinding robots equipped with vision devices or measurement of grinding force, for example, enable active recognition of the shapes of the treated surfaces and measurement of machining results. Robotic belt grinding operations optimize operating parameters in real time, based on a process knowledge model and process feedback. This breaks the limitations of traditional numerical machine tools. Especially in the case of machining shaped parts, robotic grinding is gradually replacing multi-axis CNC machine tools, and becoming an alternative method of producing such parts [10]. However, it has been noted (for example, in [11]), that there are certain limitations in the widespread use of robots in precision machining due to their relatively low accuracy and repeatability compared to CNC machine tools.

Over the last dozen or so years, the results of research on robotic grinding of complex elements have been systematically enriched, and the published literature on the subject focuses mainly on feasibility studies of grinding operations using robots, as well as modeling and analysis of machining dynamics [12]. The research is conducted in the fields of robot position optimization [13], robot calibration, grinding path planning [14,15], and material removal control by grinding force control [16,17,18].

Off-line path planning and demonstration-based grinding path learning are used in machining trajectory planning. In [19], the appropriate machining path for robotic belt grinding is assessed by trial and error. In [20], guidelines for planning the trajectory of robots are provided to the programmer by an experienced grinder. The performance and kinematic capabilities of the robot are taken into account. In some studies, automatic learning algorithms were used [21,22] on the basis of demonstrations and recording of the trajectory of manual grinding. The learning was carried out by measuring a qualified operator’s arm positions while performing the grinding process [23].

The subject of programming the robot’s work is also the subject of other works [24,25,26]. It has been shown, for example, that by controlling the grinding force, it is possible to indirectly control the material removal rate in order to improve the quality of grinding, primarily to avoid surface defects caused by improper force [27]. These results indicate that controlling the normal grinding force is one factor in improving the quality of abrasive belt sanding.

As the shape of the workpiece becomes more complex, the robot in the process of grinding the belt is prone to various types of collisions with the workpiece and tool [28]. Therefore, it is necessary to study the methods of collision-free planning of the robot’s motion path. The collision-free path planning problem in robot belt grinding is similar to the collision-free tool path planning problem in numerically controlled belt grinding, which includes local and global disruptions. A local collision means that the tool collides with the workpiece during machining. Global collision refers to the collision between the workpiece and the machine, or between the tool and the machine during machining. Global collisions are more harmful than local collisions because they can damage machines.

Another problem in belt grinding is the limited rigidity of the system: robot-tool-workpiece. This system includes, among others:Elastic deformation of the abrasive belt. On the one hand, this allows the abrasive belt to adhere closely to the surface of the workpiece. On the other hand, the steepness makes it difficult to control the grinding depth.Elastic deformation of the robot due to the open structure of the kinematic chain. As a result, the accuracy of operation and repeatability of the robot’s positioning is relatively low.

Effective solutions to the stiffness problem leading to the reduction of positioning errors are presented in [29].

Robot belt grinding can overcome the disadvantages of manual belt grinding due to its high efficiency and precision. Therefore, it has found wide application in industrial production.

As a finishing process, grinding with an abrasive belt not only makes high material removal rates possible, but can also be used to improve the surface roughness of the elements [30,31]. This is confirmed by the results of studies conducted, for example, by Hamdi [32], which show that processing with abrasive belts is an efficient and economically justified process, and provides an excellent surface finish for hardened steel, and makes a surface roughness at the level of Ra = 0.08 µm possible.

By using an industrial robot, a flexible production cell can be created that is suitable for belt grinding of parts with complex geometries. However, it should be noted that in terms of technology, grinding with abrasive tape is a complex process [33,34,35,36,37]. The material removal rate is influenced by many factors, such as belt speed, feed speed and contact pressure distribution [38,39].

The possibilities of modern industrial robots replacing humans in performance of grinding tasks have become a premise for undertaking research in this direction at Aesculap Chifa Polska, a medical tools manufacturer. The research presented in the article is innovative and covers a significant technological gap, the filling of which will stabilize the quality of surgical instruments, and in the long run reduce their production costs. The most important benefit of replacing manual grinding with robotic grinding will be relieving people from the burdensome, non-ergonomic and unsatisfactory work which should be—with the technological possibilities available today—eliminated as far as possible.

## 2. Goals, Methods and Plan of the Research

### 2.1. Goals

The aim of the research was to compare the qualitative ability of the manual belt grinding process and the robot belt grinding of the shaped surfaces of the selected surgical instrument.

The main problem with this was adopting an appropriate machining strategy that could cope with:variety of surgical tool shapessmall production serieslarge variation in shape and size deviations of the ground workpieces.

The tool shown in Figure 3, known by surgeons as orthodontic forceps, was selected for the research. In the remainder of this paper, orthodontic forceps are also referred to as workpiece or part.

The forceps are made of X20Cr13 steel. This is a martensitic chrome alloy steel that offers a good corrosion resistance and ductility, as well as a high amount of hardness. Its use on surgical instruments is indicated in ISO 7153-1:2016, Surgical instruments—materials. Part 1: Metals.

The grinding process of the nonfunctional surfaces of the forceps consists of three stages (Figure 4):I.Grinding the inner surface. The surface is curved along the axis of the workpiece, mirroring the profile of the forceps arm. In the direction perpendicular to the axis of the workpiece, the surface is flat.II.Grinding the tip of the arms that form elliptical paraboloid surfaces.III.Grinding an outer surface with curvature along the axis of the workpiece (such as an inner surface). The surface has the shape of a semi-oval in cross-section to the axis.

The criteria for comparing manual and robotic grinding were adopted taking into account the quality requirements for the machined parts. The following criteria were adopted:surface cleanliness

The presence of traces after technological operations performed on this surface before grinding is decisive for the cleanliness of the surface. In particular, these traces occur in the form of untreated surface portions (Figure 5). This may result from large unevenness of the surface after the previous treatments (forging and machining) or too small thickness of the allowance layer removed during grinding. The cleanliness of the ground surface is visible to the “naked” eye. Therefore, it was determined visually and expressed on a two-level scale (1—clean surface, 0—surface with traces of previous treatments).

dimensional repeatability

The criterion for assessing the dimensional accuracy was dimension repeatability which is determined as the dispersion of the measured values (by analogy with the repeatability in the measurement systems analysis). Practically, repeatability was determined as the difference between the smallest and the largest thickness of the arm in a selected section in the given sample of machined parts (arms).

surface roughness

Roughness was measured by the Rz parameter. Since the non-functional surfaces of the arms are subjected to an additional finishing treatment with abrasive blasting, the surface roughness after the abrasive belt treatment, which is of a shaping nature, is not limited with an exact value. Under industrial conditions it is assumed it has a value of about 10 µm. In practice, with the surface roughness slightly above 10 µm, non-functional surfaces—after treatment with loose abrasive—acquire the properties expected by end users.

shape accuracy

The shape deviation is defined as the largest deviation between the profile of the reference CAD model and the actual (after grinding) profile in the cross-section of the forceps arm. The profile of the arm after grinding was determined on the basis of a scan made with an optical scanner Atos Core 300 from GOM Metrology (Figure 6).

surface condition

Burn marks on the surface cannot occur. Two-state evaluation was used: 1—no burns, 0—traces of burns.

**Figure 5 materials-16-00630-f005:**
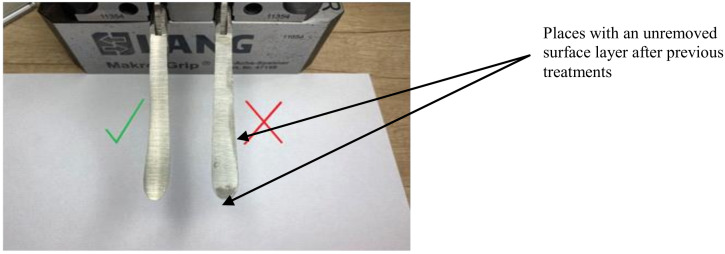
Surface cleanliness assessment.

**Figure 6 materials-16-00630-f006:**
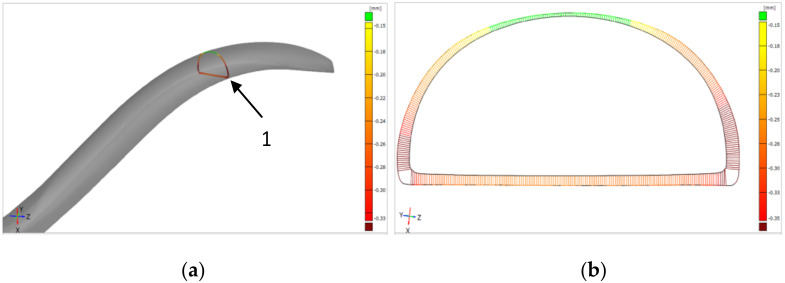
The idea of determining the shape deviation of the cross-section of the forceps arm: (**a**) location (1) of the cross-section along the length of the arm; (**b**) map of deviations between the reference profile (CAD profile) and the profile obtained after the grinding process.

### 2.2. Research Stand for Robotic Belt Grinding

The research stand for belt grinding with the use of a FANUC robot is shown in Figure 7. The elements of the stand are an industrial robot and a belt grinder. The stand is equipped with a force sensor to measure the pressure of the machined part on the abrasive belt, and a laser sensor for measuring the machined surface inclination in relation to the abrasive belt. The stand is complemented by the workpiece magazine and dust extraction.

The processed part is attached to the robot by a pneumatic holder, and performs working movements relative to the grinding belt.

The characteristics of the robot, grinder, tools and materials used in the research are presented in Table 1.

### 2.3. Machining Path Programming

The path of movement of the robot arm in relation to the grinding belt was designed in relation to the 3D model of the workpiece. Due to the curvature and convexity of the workpiece (Figure 8), surface modeling was used to create its model.

The SprutCAM Tech Ltd system (version 15) was used to design and record the path of the robot’s arm. It is CAD/CAM software used as a standard for programming numerically controlled machine tools. When generating the trajectory of the arm’s movement, all robot axes were taken into account, as well as limitations that could lead to a collision of the workpiece with the grinding belt or with robot assemblies. There is also a possibility of detecting all singularities.

The system interface was divided into three operating modes:3D model—enabling import and preparation of a geometrical model of a workpiece, including 2D drawingmachining—used to create machining processessimulation—simulation of a machining program.

When developing a strategy for grinding the surfaces indicated in Figure 6, reference was made to the experience of operators performing this work by hand. Attempts were made to reflect the movements of operators during individual treatments. Due to the limitations of the robot’s kinematics, it was necessary to create a program that takes into account both the grinding tactics used by the operators and the limitations of the robot’s kinematics.

In the case of grinding the inner surface, a strategy defined by the CAM system as contouring was used, and consisted of machining along a curve coinciding with the axis of symmetry of this surface (Figure 8).

As the arms of the forceps may be twisted axially in the operations preceding grinding, it is necessary to correct their position in relation to the grinding belt. Failure to correct it could result in an uneven removal of the material layer from the inner surface. In an extreme case, part of the surface would remain untreated. The torsion angle of the arm was determined using a laser profilometer (MICRO-EPSILON, Germany).

A 5D surface shaping was used to grind the tips and external parts of the forceps. According to this strategy, transitions are generated on the machined surface perpendicular to the defined curve and between the two indicated curves. Since the ground surfaces are convex, the contact plane of the abrasive belt is much smaller than when grinding internal parts. For this reason, any twisting of the arms does not significantly affect the grinding results.

The individual passes and strategies for grinding the outer surface and the tip of the workpiece are shown in Figure 9.

### 2.4. Grinding Parametres

Implementation of the grinding strategy on the robot described in Section 2.3 requires the following machining parameters (Figure 10):

grinding speed—*v_s_* [m/s]grinding normal force—*F_n_* [N] (the pressure of the workpiece on the abrasive belt)workpiece feed in the direction tangential to the abrasive belt—*v_f_* [mm/s]angular velocity of the workpiece rotation relative to the grinding belt—*ω_w_* [number of cycles**/**min]number of passes of the workpiece along the abrasive belt—*n*_c_

The grinding normal force *F_n_* is set indirectly by means of the so-called offset. The term offset is understood as the amount of workpiece displacement beyond its point of contact with the abrasive belt in the zero position. Its size affects the value of the pressure of the abrasive belt on the ground surface.

The parameters *v_f_*, ω_w_ and the number of passes *n_c_* were determined on the basis of the experience of the process operator and are defined in the machining program as constant values.

## 3. Results and Discussion

To determine the most favorable set of *F_n_*, *v_s_* and *v_f_* an experiment referred to as OFAT (one-factor-at-a-time method) was conducted. Three levels were adopted for each of the three parameters (in the experiment, parameters are referred to as factors) (Table 2). The range of parameters was determined on the basis of observations of grinder operators’ work during manual grinding.

According to the OFAT method, first the values of the first factor *(F_n_)* were changed with the values of the other factors remaining fixed. Next, the values of the second factor *(v_s_)* were changed, with the value of the first factor being set at the level that gave the most favorable result. The procedure was repeated until all factors were tested. For each combination of factors, *n* = 10 elements were ground.

The initial factors were established at the level of:*Fn* = 10 N*v_s_* = 26 m/s*v_f_* = 4 mm/s

The run of the experiment and the results for the initial set of factors are shown in Table 3 and Figure 11.

With the grinding force *F_n_* = 5 N, in some products the surface is not clean. The surface roughness is also high in this case.

Due to the roughness of the surface, the best results were obtained for the grinding force *F_n_* = 15 N. Such a force is sufficient to remove the entire surface layer from the surface after the previous treatment (forging, heat treatment). However, at this force the greatest dimension spread is achieved. In addition, there are traces of burns on the surface. Taking into account all the criteria, the force *F_n_* = 10 N was adopted for the next stage of the experiment. Its results are presented in Table 4 and Figure 12.

At the grinding speed v_s_ = 29 m/s, the lowest surface roughness is obtained, but on some surfaces full cleanliness is not achieved. Traces of grinding burns are also found. Following these results, the speed v_s_ = 26 m/s was adopted for the next stage of the experiment. The results of the last stage of the experiment are presented in Table 5 and Figure 13.

In the case of the workpiece feed, the best results are obtained for the feed *v_f_* = 4 mm/s. At *v_f_* = 2 mm/s, part of the surface is not 100% clean, and when *vf* = 6 mm/s, the greatest dimensional variation is obtained.

The parameters shown in Table 6 were adopted for the comparative research of manual and robotic grinding

The abrasive belts used for manual grinding showed low durability in the robotic grinding. Therefore, in robotic grinding, abrasive belts with a special base material were used. However, graininess, which is decisive for grinding accuracy and quality of the ground surface, was comparable in both types of tapes.

The grinding speed was fixed at a value of 26 m/s in the manual grinding process and in the case of robot grinding.

In manual grinding, the normal grinding force (the pressure of the workpiece on the belt) and the feed of the workpiece are selected by the operator of the process. Both are continuously and intuitively corrected by the operator. The criterion for correction is obtaining the appropriate shape and quality of the surface.

In each manual and robotic grinding test, *n* = 50 elements were ground.

The results of both manual and robotic grinding processes are presented in Table 7 and Figure 14.

In both methods, the requirements regarding the cleanliness of the surface and the absence of burns are met.

Comparable results were also obtained with the surface roughness and dimensional variability. However, when grinding with a robot, the roughness exhibits a smaller spread compared to manual grinding.

Grinding with a robot ensures a better shape accuracy in the cross-section of the arm of the ground surgical instrument (Figure 15). In the presented figure, in both cases the deviation of the shape from the nominal outline is positive. With manual grinding, it changes along the circumference of the cross-section in the range of 0.17–0.43 m; for robotic grinding it is in the range of 0.15–0.22 mm.

The explanation for this is related to the way the workpiece is guided relative to the abrasive belt. The repeatability of the tool path in the case of a grinder is limited by the physical and psychological conditions of a human. A human is not able to maintain a constant trajectory of the tool movement; moreover, a human instinctively tries to adjust movements to the irregularities of the shape of the polished profile. In the case of a robot, the path of movement is repeatable and its variability is limited only by the accuracy of the robot. In the described tests, the accuracy of the robot’s positioning was 0.03 mm (Table 1). It is practically unattainable for humans to achieve such accuracy.

## 4. Conclusions

The aim of the research was to demonstrate that manual belt grinding used in the finishing of shaped surfaces of surgical instruments can be replaced—taking into account the quality of treatment—by grinding with the use of an industrial robot. Orthodontic forceps were the surgical instrument investigated. Studies have shown that both grinding technologies are comparable, and in some respects robot-assisted grinding is superior to manual grinding. Robot grinding produces similar results in terms of surface roughness and the condition of the top layer, ensuring greater stability, and smaller errors in the shape of the cross-section of the arms of the processed forceps. It should be emphasized that the shape error is one of the most important criteria for assessing the quality of the processing of surgical instruments.

In further research, it is necessary to undertake work aimed at the possibility of controlling shape errors along the surgical tool arms, which may be the result of the robot duplicating the input shape errors after plastic processing and after the initial assembly of the tool (these errors were not controlled in the research described in the article). Therefore, the introduction of grinding force control systems or a tool path correction system, depending on the initial shape errors, is planned.

The article does not discuss the economic aspects of replacing manual belt grinding with robotic grinding. In large enterprises producing surgical instruments, the number of manual grinding stations is large (several dozen or more). Therefore, it should be taken into account that replacing them with robotic positions is associated with incurring significant costs. However, since one employee will be able to operate several robotic stations, and the optimization of robotic grinding parameters will allow for reduction of number of stations needed in comparison to manual grinding stations, it can be expected that the economic effect will be positive. Therefore, in the studies planned for the future, one of the first tasks will be to conduct research according to advanced experimental plans, allowing us to determine interactions between factors relating to grinding conditions, such as grinding parameters or grinding belt characteristics.

It should also be emphasized that, irrespective of the economic effects, the most important benefit of replacing manual grinding with robotic grinding is relieving people from the burdensome, unergonomic and unsatisfactory work, which should be eliminated as much as possible in the era of Industry 4.0.

## Figures and Tables

**Figure 1 materials-16-00630-f001:**

Examples of different shapes of surgical instruments.

**Figure 2 materials-16-00630-f002:**
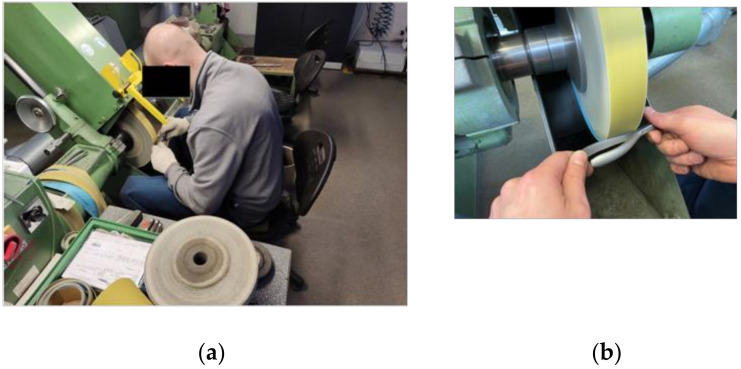
Manual belt grinding station: (**a**) overall view; (**b**) working area.

**Figure 3 materials-16-00630-f003:**
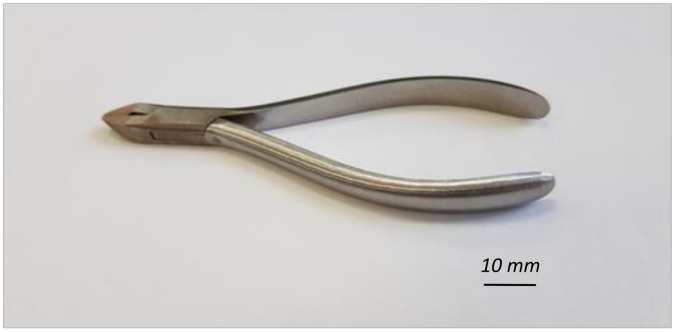
Workpiece—orthodontic forceps.

**Figure 4 materials-16-00630-f004:**
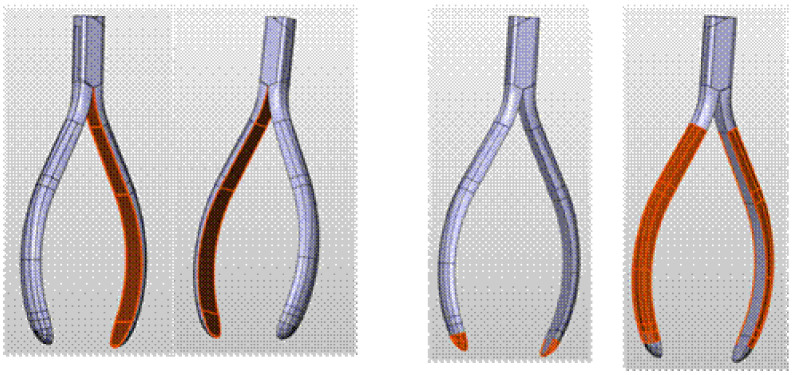
The processed surfaces of the forceps: all red marked surfaces.

**Figure 7 materials-16-00630-f007:**
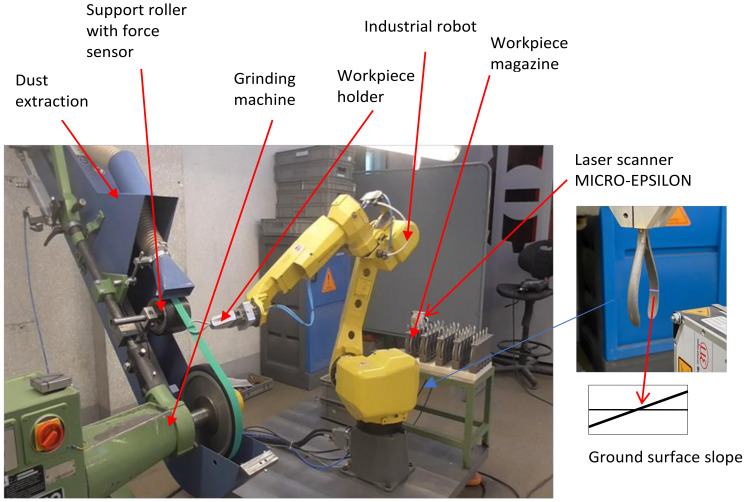
Research stand for belt grinding with the use of an industrial robot.

**Figure 8 materials-16-00630-f008:**
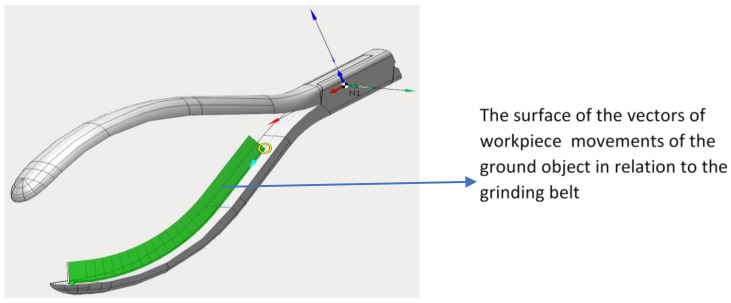
Grinding strategy of the inner surface of the workpiece.

**Figure 9 materials-16-00630-f009:**
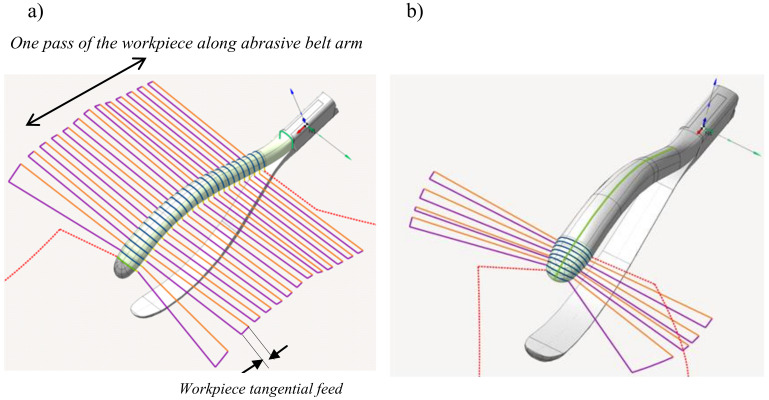
Grinding strategy: (**a**) of the outer surface; (**b**) of the tip of the workpiece.

**Figure 10 materials-16-00630-f010:**
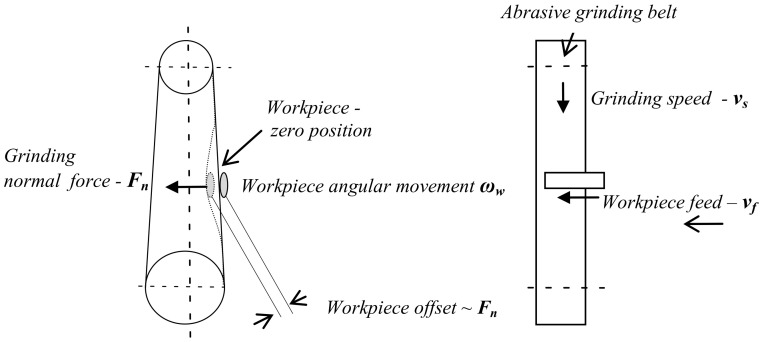
Robotic belt grinding parameters.

**Figure 11 materials-16-00630-f011:**
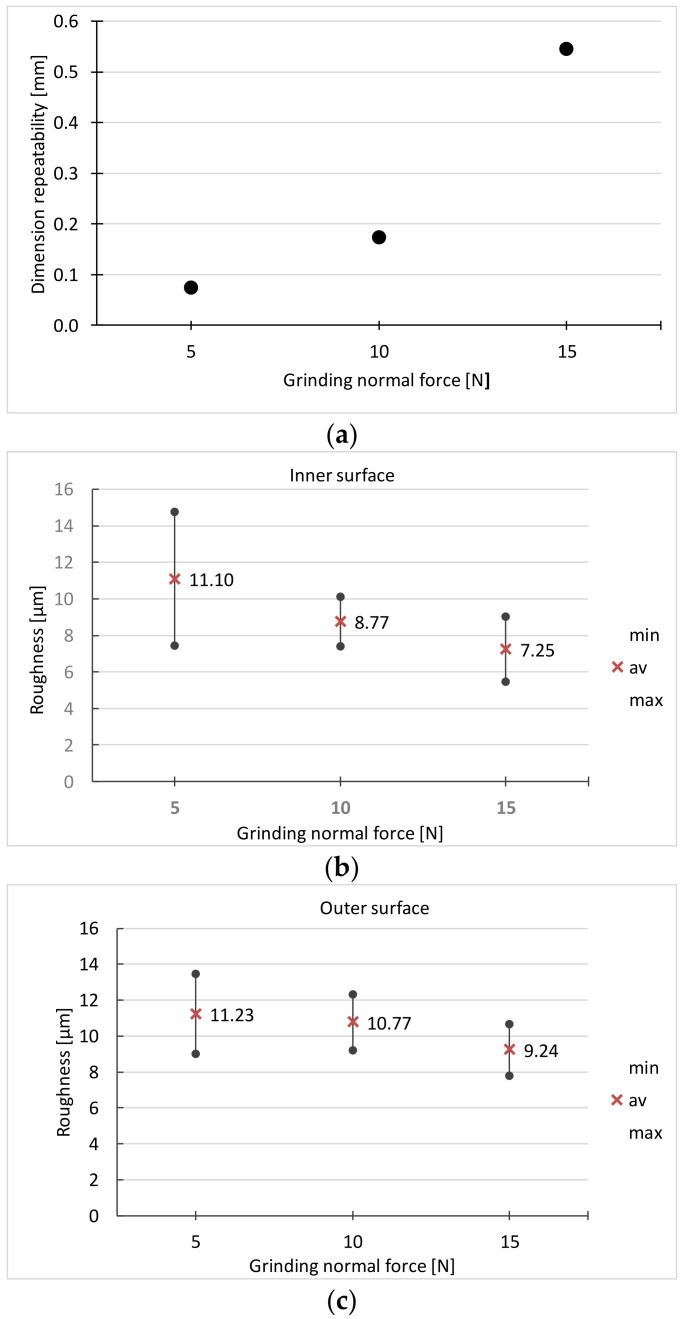
Dependence on the ginding force of: (**a**) dimensional repeatability, (**b**,**c**) surface roughness (*v_s_* = 26 m/s; *v_f_ =* 4 mm/s).

**Figure 12 materials-16-00630-f012:**
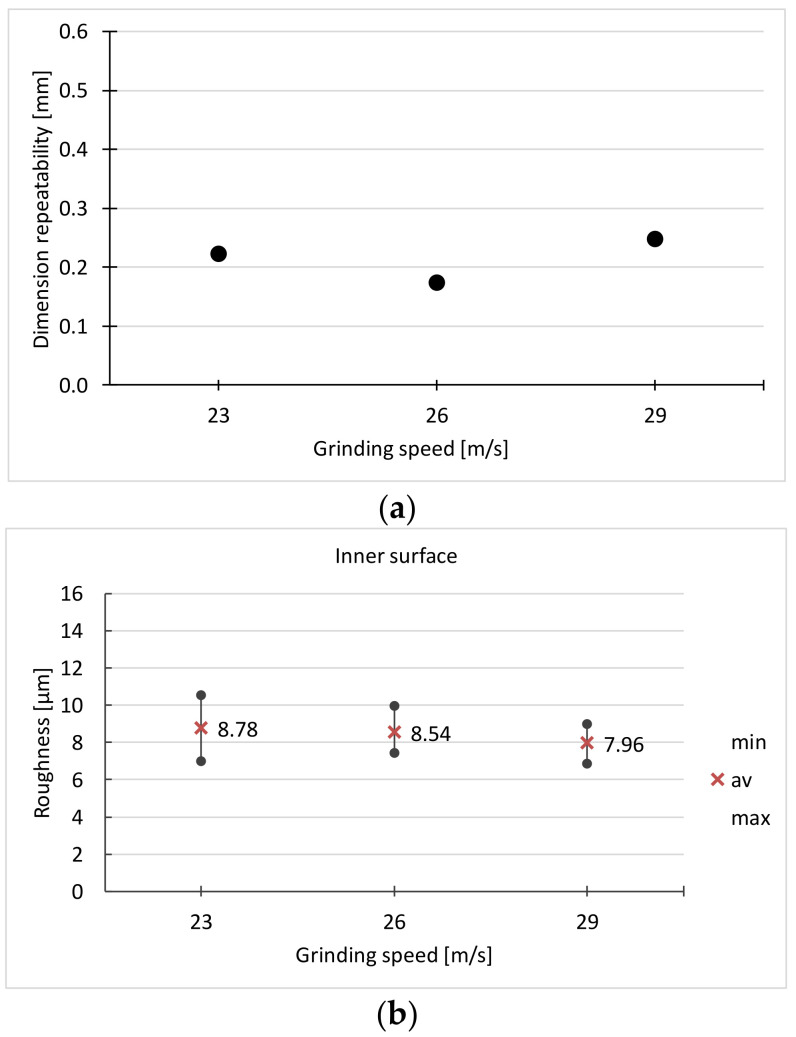
Dependence on the grinding speed of: (**a**) dimensional repeatability, (**b**,**c**) surface roughness (*F_n_* = 10 N, *v_f_* = 4 mm/s).

**Figure 13 materials-16-00630-f013:**
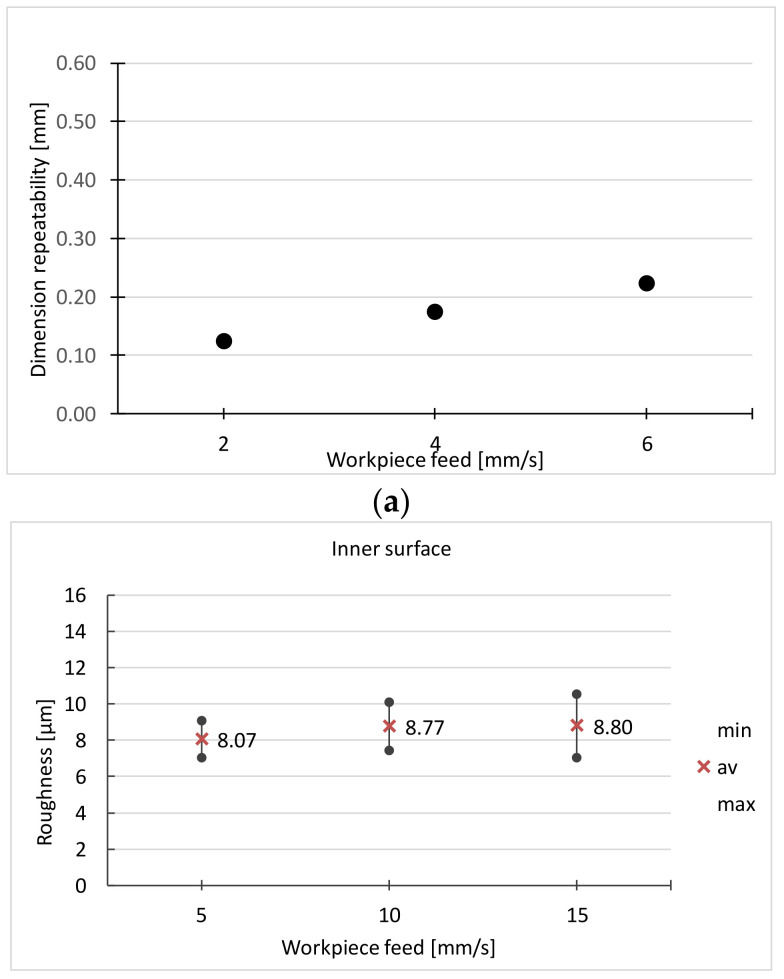
Dependence on the workpiece feed of: (**a**) dimensional repeatability, (**b**,**c**) surface roughness (*F_n_* = 10 N, *v_s_* = 26 m/s).

**Figure 14 materials-16-00630-f014:**
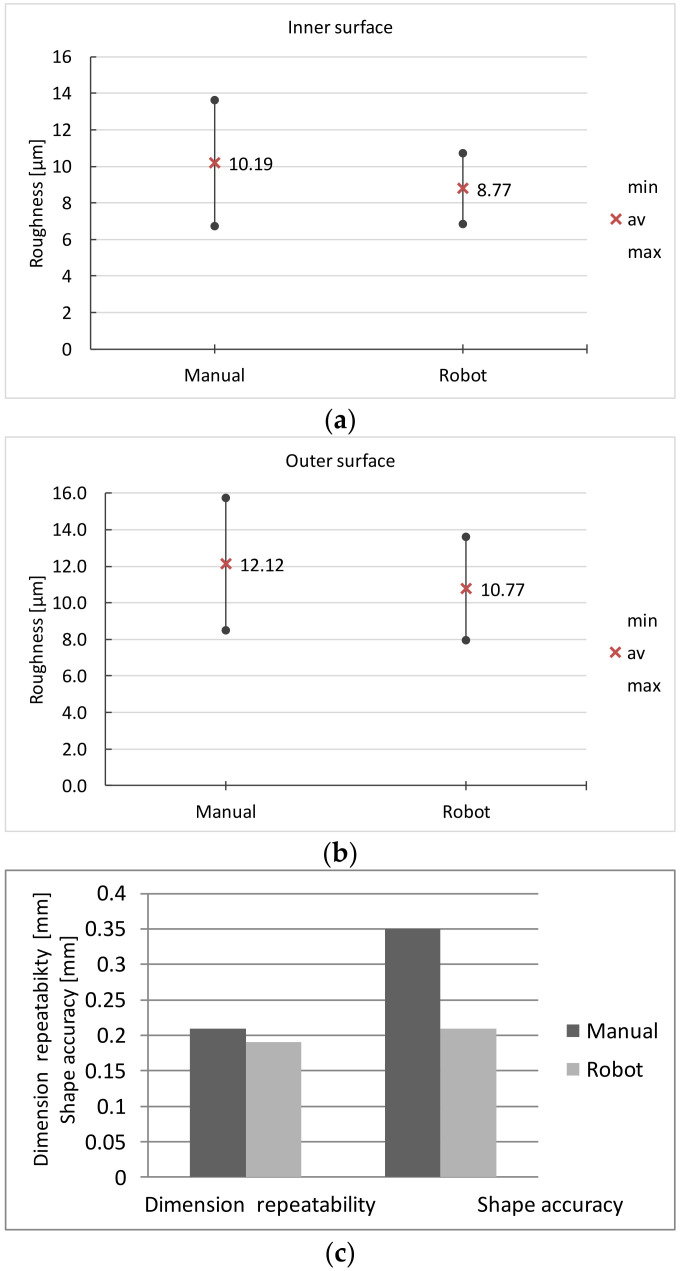
Comparison of: (**a,b**) surface roughness and (**c**) dimensional repeatability and shape accuracy obtained with manual and robot grinding.

**Figure 15 materials-16-00630-f015:**
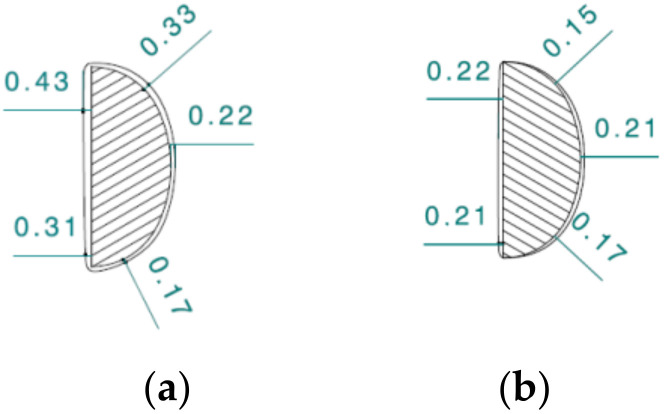
Shape error in cross-section of the forceps arm after: (**a**) manual grinding and (**b**) grinding with the use of robot.

**Table 1 materials-16-00630-t001:** The characteristics of the robot and the grinder.

Robot	
Type	FANUC M-10iA/10M (high inertia version)
Controlled axes	6
Repeatability	0.03 mm
Mechanical weight	130 kg
Maximum load capacity at wrist	10 kg
Motion range	360^o^
Maximum speed	360^o^/s
Grinder	
Type	Meatabo
Electric motor power	3 kW

**Table 2 materials-16-00630-t002:** Plan of the experiment.

Factor	Factor Level		
	1	2	3
*F_n_*	5 N	10 N	15 N
*v_s_*	23 m/s	26 m/s	29 m/s
*v_t_*	2 mm/s	4 mm/s	6 mm/s

**Table 3 materials-16-00630-t003:** Grinding quality depending on grinding normal force setting (*v_s_* = 26 m/s; *v_f_* = 4 mm/s).

Grinding Normal Force -*F_n_* [N]	Surface Clean- Liness	Dimensional Repeatability [mm]	Roughness Inner Surface [µm]		Roughness Outer Surface [µm]		Surface Condition
		Mean	Mean	Range	Mean	Range	
5	0	0.07	11.1	7.14	11.23	4.44	1
10	1	0.17	8.77	2.70	10.77	3.12	1
15	1	0.55	7.25	3.60	9.24	2.88	0

**Table 4 materials-16-00630-t004:** Grinding quality depending on grinding speed (*F_n_* = 10 N, *v_f_* = 4 mm/s).

Grinding Speed - *v_s_* [m/s]	Surface Clean- Liness	Dimensional Repeatability [mm]	Roughness Inner Surface [µm]		Roughness Outer Surface [µm]		Surface Condition
		Mean	Mean	Range	Mean	Range	
23	1	0.22	8.78	3.70	11.13	3.74	1
26	1	0.17	8.54	2.70	10.77	3.12	1
29	0	0.25	7.96	1.76	9.35	1.07	0

**Table 5 materials-16-00630-t005:** Grinding quality depending on the feed of the workpiece (*F_n_* = 10 N, v_s_ = 26 m/s).

Feed - *v_f_* [mm/s]	Surface Clean- Liness	Dimensional Repeatability [mm]	Roughness Inner Surface [µm]		Roughness Outer Surface [µm]		Surface Condition
		Mean	Mean	Range	Mean	Range	
2	0	0.12	8.07	2.03	10.65	5.36	1
4	1	0.17	8.77	2.70	10.77	3.12	1
6	1	0.22	8.8	3.52	10.59	2.80	1

**Table 6 materials-16-00630-t006:** Grinding parameters used in comparative research.

	Robotic	Manual
Grinder	Metabo	Metabo
Grinding Belt	Klingspor LS312 J-Flex P150	Hermes Ceramit CN 466 X-Flex 150
Grinding speed—*v_s_*	26 m/s	26 m/s
Grinding normal force—*F_n_*	10 N	Determined by the grinder
Workpiece feed—*v_f_*	4 mm/s	Determined by the grinder
Sample size	50	50

**Table 7 materials-16-00630-t007:** The results of manual and robotic belt grinding.

Grinding Method	Surface Clean- Liness	Dimensional Repeatability [mm]	Roughness Inner Surface [µm]		Roughness Outer Surface [µm]		Surface Condi- tion	Shape Accuracy
		Mean	Mean	Range	Mean	Range		Mean
Manual	1	0.21	10.19	6.89	12.12	7.21	1	0.35
With robot	1	0.19	8.77	4.00	10.77	6.28	1	0.21

## Data Availability

Not applicable.

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
