# Peer review of "The Quality of Surgical Instrument Surfaces Machined with Robotic Belt Grinding"

_materials, 2023, doi:10.3390/ma16020630_

Round 1

Reviewer 1 Report

This paper discusses the quality of surgical instrument surfaces by robotic belt grinding, and provides the comparison between manual and robotic grinding. The work has certain practical meanings. The suggestions are

1.     The cleanliness should be defined.

2.     What is the cost of robotic belt grinding compared to manual methods?

3.     The roughness decreases with grinding normal force in the given range. I wonder what can be the lower limitation of roughness obtained under an extreme force?

4.     The roughness of inner surface is non-monotonous with speed, what is the reason?

Reviewer 2 Report

1. The manuscript has been written like a report, not like a scientific finding. 

1. The discussion part is not written well.

2. All the figures are not explained properly. Most of the figures and plots should be sub-sectioned, like Fig. 1(a), (b), etc. and should be explained each sub-section of the figure.

3 Conclusion is missing.

Reviewer 3 Report

- Fig. 3 - missing scale (dimensions of the tool)

- it should be considered to describe in the title or/and in the manucript abstract, that it deals with "non - fuctional surfaces" of the surgical tools 

- Tab. 3 - roughness of outer tool surface is 10,77 microns for winning  grinding normal force Fn - is it okey? You defined in text above, that the limit is 10 microns. How did you measure and evaluate roughness parameter?

- it would be good and also for readers comfort to show the surface roughness limit as well as dimension repeatability in all graphs within Fig. 11 - 14 /display a treshold line at 10 microns/

- experiment was planned according to DoE - why you did not apply other method for evaluation like Pareto chart or others (correlation matrix could be very useful also) to describe realtionship among input and output parametres 

- surface plots could also describe a functional relationship between two independent variable - why they were not used? 

- comparative research between manual/robot grinding was perfomed, but with different grinding belt? It is not comparable any more...

- Fig. 7 - Laser sensor (small figure botom right) is unclear and unreadable 

Reviewer 4 Report

This paper investigates the performance of a robotic belt grinding operation comparing with a manual belt grinding operation conducting by human operator. Authors initially conducted some experiments to find optimal grinding parameters (grinding normal force, grinding speed, and workpiece feed) which provide the best performance. However, the authors did not consider the interactions among the cutting parameters as they applied one-factor-at-a-time method. Therefore, it cannot be claimed that these grinding parameters present the best performance. This can be seen okay as the manuscript does not investigate the best grinding parameters for a robotic grinding operation.

The robotic grinding with 'the best' parameters was compared with the manual grinding but the problem here is that the manual grinding operation is not repeatable as it depends on the skill and experience of an operator (or more than one-this is not stated in the manuscript. ). This does not follow the scientific approach. Also, it raises a question: Can a more skilled or experienced operator outperform a robotic grinding? Because the performance differences in roughness for outer surface and dimension repeatability are not significantly large.

Moreover, the structure of the manuscript is overall not appropriate. The introduction suffers from presenting the gap and novelty of the paper. There is no conclusion. The paper must be structured and written better. In view of this reviewer, the manuscript with its current version is not suitable for publication for this journal. The study can be improved with the following comments:

1) The introduction presents only previous studies. It must also present the novelty and the gap addressed in the submitted study, after evaluating the previous studies in the literature. This is missing in the introduction.

2) p.4 Line150 - More information is needed, such as its material, why was it chosen? Is this because of its complex shape? Other reasons should be stated if any exists.

3) It can be good if each subfigure in Figure 4 is marked as (a), (b),... and the grinding process (Line 154-161) refers to these subfigures.

4) Line 173 - How was the surface cleanliness measured or decided? By eyes?

5) Line 175 - How was the dimensional repeatability measured?

6) Images in Figure 6 are not clear. There is an orange mark in the first subfigure. That is not clear. In the second subfigure, there is line which indicates nothing. Both subfigures need to be amended.

7)Line 282 - One-factor-a-time-method was applied. If the interactions of the parameters are considered, there is a possibility of that a better grinding parameters can be obtained.

8) Line 292, '...  'n=10 elements' is not clear to me. 'n' is not defined. Is this the number of samples ground?  

9)The comparison between the robotic and manual grinding highly depends on the operator skill and experience, which has not been mentioned in the text. If the results depends on human's skill and experience, more details should be provided about it to eliminate possible questions. 

10) Table 6 - Determined by the grinder for robotic grinding? The table should follow Line 364.

11) The manuscript must have a conclusion section. 

12)Minor comments:

Line 251- dot, instead of '/'

Line - '[', instead of '('

Line 279 - 'w' and 'c' should be subscript.

Table 2 - v_f instead of v_t

Line 385 - 'better' instead of 'batter'

Round 2

Reviewer 2 Report

I think now the manuscript is good to accept for the journal in this revised form.

Author Response

Dear Reviewer,

 Thank you for accepting our explanations and amendments.

 Authors

Reviewer 4 Report

I thank the authors to address the issues that were stated in the comments. There are a few points that need to be addressed to improve the manuscript:

1) The novelty and gap must be stated more specifically. What is written in Conclusion is good. This can be rephrase in Introduction. It would be good if it is located close to the end of Introduction after presenting the literature works.

2)The quality of Figure 6 is still not good. Possibly, the white rectangular boxes are still can be seen. Some parts of text are missing (please see "section after grinding").

3) The authors stated "...It was considered that the parameters of grinding with a robot should be close to the best - in given conditions. To demonstrate this, an OFAT type experiment is sufficient.

In further studies, studies will be carried out according to more advanced plans, allowing to determine interactions between factors.". It would be good if this can be added into Conclusion as future work.

Author Response

Dear Reviewer,

Thank you for accepting our explanations and amendments.

Below are the replies to the next three remarks:

  • The novelty and gap must be stated more specifically. What is written in Conclusion is good. This can be rephrase in Introduction. It would be good if it is located close to the end of Introduction after presenting the literature works.

The presentation of novelty (innovation) and research gap has been strengthened and clarified. The relevant fragments have been added at the end of the introduction (lines 139-147).

2)The quality of Figure 6 is still not good. Possibly, the white rectangular boxes are still can be seen. Some parts of text are missing (please see "section after grinding").

In Figure 6, the descriptions on the profiles, which could be misleading, have been omitted. The description is included in the caption under the drawing (lines215-218). Figure 6 shows only the idea of determining the shape deviation. For better explanation, a short description of the measurement has been added (line 204-207).

3) The authors stated "...It was considered that the parameters of grinding with a robot should be close to the best - in given conditions. To demonstrate this, an OFAT type experiment is sufficient.

In further studies, studies will be carried out according to more advanced plans, allowing to determine interactions between factors.". It would be good if this can be added into Conclusion as future work.

Information about future studies using more advanced experiment plans has been added (lines 463-466).

Authors